# One-step differentiation of iterative algorithms

**Jérôme Bolte**
Toulouse School
of Economics,
Université Toulouse
Capitole,
Toulouse, France.

**Edouard Pauwels**
Toulouse School
of Economics (IUF),
Toulouse, France.

**Samuel Vaiter**
CNRS &
Université Côte d'Azur,
Laboratoire J. A. Dieudonné.
Nice, France.

## Abstract

In appropriate frameworks, automatic differentiation is transparent to the user at the cost of being a significant computational burden when the number of operations is large. For iterative algorithms, implicit differentiation alleviates this issue but requires custom implementation of Jacobian evaluation. In this paper, we study one-step differentiation, also known as Jacobian-free backpropagation, a method as easy as automatic differentiation and as efficient as implicit differentiation for fast algorithms (e.g., superlinear optimization methods). We provide a complete theoretical approximation analysis with specific examples (Newton's method, gradient descent) along with its consequences in bilevel optimization. Several numerical examples illustrate the well-foundedness of the one-step estimator.

## 1 Introduction

Differentiating the solution of a machine learning problem is an important task, e.g., in hyperparameters optimization [9], in neural architecture search [31] and when using convex layers [3]. There are two main ways to achieve this goal: *automatic differentiation* (AD) and *implicit differentiation* (ID). Automatic differentiation implements the idea of evaluating derivatives through the compositional rules of differential calculus in a user-transparent way. It is a mature concept [27] implemented in several machine learning frameworks [36, 16, 1]. However, the time and memory complexity incurred may become prohibitive as soon as the computational graph becomes bigger, a typical example being unrolling iterative optimization algorithms such as gradient descent [5]. The alternative, implicit differentiation, is not always accessible: it does not solely rely on the compositional rules of differential calculus (Jacobian multiplication) and usually requires solving a linear system. The user needs to implement custom rules in an automatic differentiation framework (as done, for example, in [4]) or use dedicated libraries such as [11, 3, 10] implementing these rules for given models. Provided that the implementation is carefully done, this is most of the time the gold standard for the task of differentiating problem solutions.

**Contributions.** We study a *one-step Jacobian* approximator based on a simple principle: differentiate only the last iteration of an iterative process and drop previous derivatives. The idea of dropping derivatives or single-step differentiation was explored, for example, in [24, 23, 39, 22, 40, 29] and our main contribution is a general account and approximation analysis for Jacobians of iterative algorithms. One-step estimation constitutes a rough approximation at first sight and our motivation to study is its ease-of-use within automatic differentiation frameworks: no custom Jacobian-vector products or vector-Jacobian products needed as long as a `stop_gradient` primitive is available (see Table 1).

37th Conference on Neural Information Processing Systems (NeurIPS 2023).

| Algorithm 1: Automatic | Algorithm 2: Implicit | Algorithm 3: One-step |
|---|---|---|
| **Input:** $\theta \mapsto x_0(\theta) \in \mathcal{X}$, $k > 0$. **Eval:**   with_gradient   **for** $i = 1, \ldots, k$ **do**    $x_i(\theta) = F(x_{i-1}(\theta), \theta)$   **Return:** $x_k(\theta)$ **Differentiation:** native autodiff on **Eval**. | **Input:** $x_0 \in \mathcal{X}$, $k > 0$. **Eval:**   stop_gradient   **for** $i = 1, \ldots, k$ **do**    $x_i = F(x_{i-1}, \theta)$   **Return:** $x_k$ **Differentiation:** Custom implicit VJP / JVP. | **Input:** $x_0 \in \mathcal{X}$, $k > 0$. **Eval:**   stop_gradient   **for** $i = 1, \ldots, k - 1$ **do**    $x_i = F(x_{i-1}, \theta)$   with_gradient   $x_k(\theta) = F(x_{k-1}, \theta)$   **Return:** $x_k(\theta)$ **Differentiation:** native autodiff on **Eval** |

Table 1: Qualitative comparison of differentiation strategies. Native autodiff refers to widespread primitives in differentiable programming (e.g. `grad` in JAX). Custom JVP/VJP refers to specialized libraries such as `jaxopt` [12] or `qpth` [4] implicit differentiation in specific contexts.

| Algorithm | Implementation | Efficient |
|---|---|---|
| **Automatic** | Native autodiff | no |
| **Implicit** | Custom | yes |
| **One step** | Native autodiff | yes |

We conduct an approximation analysis of one-step Jacobian (Corollary 1). The distance to the true Jacobian are produced by the distance to the solution (i.e., the quality of completion of the optimization phase) and the lack of contractivity of the iteration mapping. This imprecision has to be balanced with the ease of implementation of the one-step Jacobian. This suggests that one-step differentiation is efficient for small contraction factors, which corresponds to fast algorithms. We indeed show that one-step Jacobian is asymptotically correct for super-linearly convergent algorithms (Corollary 2) and provide similar approximation rate as implicit differentiation for *quadratically convergent algorithms* (Corollary 3). We exemplify these results with hypergradients in bilevel optimization, conduct a detailed complexity analysis and highlight in Corollary 4 the estimation of approximate critical points. Finally, numerical illustrations are provided to show the practicality of the method on logistic regression using Newton's algorithm, interior point solver for quadratic programming and weighted ridge regression using gradient descent.

**Related works.** Automatic differentiation [27] was first proposed in the forward mode in [44] and its reverse mode in [30]. The study of the behaviour of differentiating iterative procedure by automatic differentiation was first analyzed in [25] and [8] in the optimization community. It was studied in the machine learning community for smooth methods [35, 32, 37], and nonsmooth methods [14]. Implicit differentiation is a recent highlight in machine learning. It was shown to be a good way to estimate the Jacobian of problem solutions, for deep equilibrium network [5], optimal transport [33], and also for nonsmooth problems [13], such as sparse models [10]. Inexact implicit differentiation was explored in [19]. Truncated estimation of the "Neumann series" for the implicit differentiation is routinely used [32, 34] and truncated backpropagation was investigated in [42] for the gradient descent iterations. In the very specific case of min-min problems, [2] studied the speed of convergence of automatic, implicit, and analytical differentiation.

The closest work to ours is [22] – under the name *Jacobian-free backpropagation* – but differs significantly in the following ways. Their focus is on single implicit layer networks, and guarantees are qualitative (descent direction of [22, Theorem 3.1]). In contrast, we provide quantitative results on abstract fixed points and applicable to any architecture. The idea of "dropping derivatives" was proposed in [21] for meta-learning, one-step differentiation was also investigated to train Transformer architectures [23], and to solve bilevel problems with quasi-Newtons methods [39].

## 2 One-step differentiation

### 2.1 Automatic, implicit and one-step differentiation

Throughout the text $F \colon \mathbb{R}^n \times \mathbb{R}^m \to \mathbb{R}^n$ denotes a recursive algorithmic map from $\mathbb{R}^n$ to $\mathbb{R}^n$ with $m$ parameters. For any $\theta \in \mathbb{R}^m$, we write $F_\theta \colon x \mapsto F(x, \theta)$ and let $F_\theta^k$ denote $k$ recursive composition of $F_\theta$, for $k \in \mathbb{N}$. The map $F_\theta$ defines a recursive algorithm as follows

$$x_0(\theta) \in \mathbb{R}^n \quad \text{and} \quad x_{k+1}(\theta) = F(x_k(\theta), \theta), \tag{1}$$

We denote by $J_x F_\theta$ the Jacobian matrix with respect to the variable $x$. The following assumption is sufficient to ensure a non degenerate asymptotic behavior.

**Assumption 1 (Contraction)** Let $F \colon \mathbb{R}^n \times \mathbb{R}^m \to \mathbb{R}^n$ be $C^1$, $0 \le \rho < 1$, and $\mathcal{X} \subset \mathbb{R}^n$ be nonempty convex closed, such that for any $\theta \in \mathbb{R}^m$, $F_\theta(\mathcal{X}) \subset \mathcal{X}$ and $\|J_x F_\theta\|_{\mathrm{op}} \le \rho$.

**Remark 1** The main algorithms considered in this paper fall in the scope of smooth optimization. The algorithmic map $F_\theta$ is associated to a smooth parametric optimization problem given by $f \colon \mathbb{R}^n \times \mathbb{R}^m \to \mathbb{R}$ such that $f_\theta \colon x \mapsto f(x, \theta)$ is strongly convex, uniformly in $\theta$. Two examples of algorithmic maps are given by gradient descent, $F_\theta(x) = x - \alpha \nabla f_\theta(x)$, or Newton's $F_\theta(x) = x - \alpha \nabla^2 f_\theta(x)^{-1} \nabla f_\theta(x)$ for positive step $\alpha > 0$. For small step sizes, gradient descent provides a contraction and Newton's method provides a local contraction, both fall in the scope of Assumption 1. Other examples include inertial algorithms such as the Heavy Ball method [38], which has to be considered in phase space and for which a single iteration is not contracting, but a large number of iteration is (see *e.g.* [38, 35]).

The following lemma gathers known properties regarding the fixed point of $F_\theta$, denoted by $\bar{x}(\theta)$ and for which we will be interested in estimating derivatives.

**Lemma 1** *Under Assumption 1, for each $\theta$ in $\mathbb{R}^m$ there is a unique fixed point of $F_\theta$ in $\mathcal{X}$ denoted by $\bar{x}(\theta)$, which is a $C^1$ function of $\theta$. Furthermore, for all $k \in \mathbb{N}$, we have $\|x_k(\theta) - \bar{x}(\theta)\| \le \rho^k \|x_0(\theta) - \bar{x}(\theta)\| \le \rho^k \frac{\|x_0 - F_\theta(x_0)\|}{1 - \rho}$.*

This is well known, we briefly sketch the proof. The mapping $F_\theta$ is a $\rho$ contraction on $\mathcal{X}$ – use the convexity of $\mathcal{X}$ and the intermediate value theorem. Banach fixed point theorem ensures existence and uniqueness, differentiability is due to the implicit function theorem. The convergence rate is classical. We are interested in the numerical evaluation of the Jacobian $J_\theta \bar{x}(\theta)$, thus well-defined under Assumption 1.

The automatic differentiation estimator $J^{\mathrm{AD}} x_k(\theta) = J_\theta x_k(\theta)$ propagates the derivatives (either in a forward or reverse mode) through iterations based on the piggyback recursion [26], for $i = 1, \ldots, k-1$,

$$J_\theta x_{i+1}(\theta) = J_x F(x_i(\theta), \theta) J_\theta x_i(\theta) + J_\theta F(x_i(\theta), \theta). \tag{2}$$

Under assumption 1 we have $J^{\mathrm{AD}} x_k(\theta) \to J_\theta \bar{x}(\theta)$ and the convergence is asymptotically linear [25, 8, 35, 41, 14]. $J^{\mathrm{AD}} x_k(\theta)$ is available in differentiable programming framework implementing common primitives such as backpropagation.

The implicit differentiation estimator $J^{\mathrm{ID}} x_k(\theta)$ is given by application of the implicit function theorem using $x_k$ as a surrogate for the fixed point $\bar{x}$,

$$J^{\mathrm{ID}} x_k(\theta) = (I - J_x F(x_k(\theta), \theta))^{-1} J_\theta F(x_k(\theta), \theta). \tag{3}$$

By continuity of the derivatives of $F$, we also have $J^{\mathrm{ID}} x_k(\theta) \to J_\theta \bar{x}(\theta)$ as $k \to \infty$, (see *e.g.* [27, Lemma 15.1] or [12, Theorem 1]). Implementing $J^{\mathrm{ID}} x_k(\theta)$ requires either manual implementation or dedicated techniques or libraries [4, 3, 45, 28, 20, 12] as the matrix inversion operation is not directly expressed using common differentiable programming primitives. A related estimator is the Inexact Automatic Differentiation (IAD) estimator which implements (2) but with Jacobians evaluated at the last iterates $x_k$, which can be seen as an approximation of $J^{\mathrm{ID}}$ [35, 19].

The one-step estimator $J^{\mathrm{OS}} x_k(\theta)$ is the Jacobian of the fixed point map for the last iteration

$$J^{\mathrm{OS}} x_k(\theta) = J_\theta F(x_{k-1}(\theta), \theta). \tag{4}$$

```
def F(x, theta):                              # Implicit differentiation
    # here a gradient step                    J_id = # Custom implementation (e.g. jaxopt)
    return x - alpha * grad_f(x, theta)
                                              # One-step differentiation
                                              def iterative_procedure_with_stop_grad(theta, x0):
def iterative_procedure(theta, x0):               x = x0
    x = x0                                        for _ in range(99):
    for _ in range(100):                              x = F(stop_gradient(x),
        x = F(x, theta)                                       stop_gradient(theta))
    return x                                      x = F(x, theta)
                                                  return x
# Automatic differentiation                   J_os = jacfwd(iterative_procedure)(theta, x0)
J_ad = jacfwd(iterative_procedure)(theta, x0)
```

Figure 1: Implementation of Algorithms 1, 2 and 3 in `jax`. $F$ is a gradient step of some function $f$. The custom implementation of implicit differentiation is not explicitly stated. The function `stop_gradient` is present in `jax.lax` and `jacfwd` computes the full Jacobian using forward-mode AD.

Contrary to automatic differentiation or implicit differentiation estimates, we do not have $J^{\mathrm{OS}}x_k(\theta) \to J_\theta \bar{x}(\theta)$ in general as $k \to \infty$, but we will see that the error is essentially proportional to $\rho$, and thus negligible for fast algorithms for which the estimate is accurate.

From a practical viewpoint, the three estimators $J^{\mathrm{AD}}$, $J^{\mathrm{ID}}$ and $J^{\mathrm{OS}}$ are implemented in a differentiable programming framework, such as `jax`, thanks to a primitive `stop_gradient`, as illustrated by Algorithms 1, 2 and 3. The computational effect of the `stop_gradient` primitive is to replace the actual Jacobian $J_x F(x_i(\theta), \theta)$ by zero for chosen iterations $i \in \{1, \ldots, k\}$. Using it for all iterations except the last one, allows one to implement $J^{\mathrm{OS}}$ in (4) using Algorithm 3. This illustrates the main interest of the one-step estimator: it can be implemented using any differentiable programming framework which provides a `stop_gradient` primitive and does not require custom implementation of implicit differentiation. Figure 1 illustrates an implementation in `jax` for gradient descent.

## 2.2 Approximation analysis of one step differentiation for linearly convergent algorithms

The following lemma is elementary. It describes the main mathematical mechanism at stake behind our analysis of one-step differentiation.

**Lemma 2** *Let $A \in \mathbb{R}^{n \times n}$ with $\|A\|_{\mathrm{op}} \leq \rho < 1$ and $B, \tilde{B} \in \mathbb{R}^{n \times m}$, then*

$$(I - A)^{-1}B - \tilde{B} = A(I - A)^{-1}B + B - \tilde{B}.$$

*Moreover, we have the following estimate,*

$$\|(I - A)^{-1}B - \tilde{B}\|_{\mathrm{op}} \leq \frac{\rho}{1 - \rho}\|B\|_{\mathrm{op}} + \|B - \tilde{B}\|_{\mathrm{op}}.$$

**Proof :** First for any $v \in \mathbb{R}^n$, we have $\|(I - A)v\| \geq \|Iv\| - \|Av\| \geq (1 - \rho)\|v\|$, which shows that $I - A$ is invertible (the kernel of $I - A$ is trivial). We also deduce that $\|(I - A)^{-1}\|_{\mathrm{op}} \leq 1/(1 - \rho)$. Second, we have $(I - A)^{-1} - I = A(I - A)^{-1}$, since $((I - A)^{-1} - I)(I - A) = A$, and therefore

$$A(I - A)^{-1}B + B - \tilde{B} = ((I - A)^{-1} - I)B + B - \tilde{B} = (I - A)^{-1}B - \tilde{B}.$$

The norm bound follows using the submultiplicativity of operator norm, the triangular inequality and the fact that $\|A\|_{\mathrm{op}} \leq \rho$ and $\|(I - A)^{-1}\|_{\mathrm{op}} \leq 1/(1 - \rho)$. $\qquad\square$

**Corollary 1** *Let $F$ and $\mathcal{X}$ be as in Assumption 1 such that $\theta \mapsto F(x, \theta)$ is $L_F$ Lipschitz and $x \mapsto J_\theta F(x, \theta)$ is $L_J$ Lipschitz (in operator norm) for all $x \in \mathbb{R}^n$. Then, for all $\theta \in \mathbb{R}^m$,*

$$\|J^{\mathrm{OS}}x_k(\theta) - J_\theta \bar{x}(\theta)\|_{\mathrm{op}} \leq \frac{\rho L_F}{1 - \rho} + L_J\|x_{k-1} - \bar{x}(\theta)\|. \tag{5}$$

**Proof :** The result follows from Lemma 2 with $A = J_x F(\bar{x}(\theta), \theta)$, $B = J_\theta F(\bar{x}(\theta), \theta)$ and $\tilde{B} = J_\theta F(x_{k-1}, \theta)$ using the fact that $\|B\|_{\mathrm{op}} \leq L_F$ and $\|B - \tilde{B}\|_{\mathrm{op}} \leq L_J\|\bar{x}(\theta) - x_{k-1}\|$. $\qquad\square$

**Remark 2 (Comparison with implicit differentiation)** In [12] a similar bound is described for $J^{\mathrm{ID}}$, roughly under the assumption that $x \to F(x, \theta)$ also has $L_J$ Lipschitz Jacobian, one has

$$\|J^{\mathrm{ID}}x_k(\theta) - J_\theta \bar{x}(\theta)\|_{\mathrm{op}} \leq \frac{L_J L_F}{(1-\rho)^2}\|x_k - \bar{x}(\theta)\| + \frac{L_J}{1-\rho}\|x_k - \bar{x}(\theta)\|. \tag{6}$$

For small $\rho$ and large $k$, the main difference between the two bounds (5) and (6) lies in their first term which is of the same order whenever $\rho$ and $L_J\|\bar{x} - x_{k-1}\|$ are of the same order.

Corollary 1 provides a bound on $\|J^{\mathrm{OS}}x_k(\theta) - J_\theta\bar{x}(\theta)\|_{\mathrm{op}}$ which is asymptotically proportional to $\rho$. This means that for fast linearly convergent algorithms, meaning $\rho \ll 1$, one-step differentiation provides a good approximation of the actual derivative. Besides, given $F$ which satisfies Assumption 1, with a given $\rho < 1$, not specially small, one can set $\tilde{F}_\theta = F_\theta^K$ for some $K \in \mathbb{N}$. In this case, $\tilde{F}$ satisfies assumption 1 with $\tilde{\rho} = \rho^K$ and the one-step estimator in (4) applied to $\tilde{F}$ becomes a $K$-steps estimator on $F$ itself, we only differentiate through the $K$ last steps of the algorithm which amounts to truncated backpropagation [42].

**Example 1 (Gradient descent)** Let $f\colon \mathbb{R}^n \times \mathbb{R}^m \to \mathbb{R}$ be such that $f(\cdot, \theta)$ is $\mu$-strongly convex ($\mu > 0$) with $L$ Lipschitz gradient for all $\theta \in \mathbb{R}^m$, then the gradient mapping $F\colon (x, \theta) \mapsto x - \alpha\nabla_x f(x, \theta)$ satisfies Assumption 1 with $\rho = \max\{1 - \alpha\mu, \alpha L - 1\}$, smaller than 1 as long as $0 < \alpha < 2/L$. The optimal $\alpha = 2/(L + \mu)$ leads to a contraction factor $\rho = 1 - 2\mu/(L + \mu)$. Assuming that $\nabla^2_{x\theta}f$ is also $L$ Lipschitz, Corollary 1 holds with $L_F = L_J = 2L/(\mu + L) \leq 2$. For step size $1/L$, Corollary 1 holds with $L_F = L_J \leq 1$ and $\rho = 1 - \mu/L$. In both cases, the contraction factor $\rho$ is close to 0 when $L/\mu \simeq 1$, for well conditioned problems. As outlined above, we may consider $\tilde{F}_\theta = F_\theta^K$ in which case Corollary 1 applies with a smaller value of $\rho$, recovering the result of [42, Proposition 3.1]

## 2.3 Superlinear and quadratic algorithms

The one-step Jacobian estimator in (4) as implemented in Algorithm 3 is obviously not an exact estimator in the sense that one does not necessarily have $J^{\mathrm{OS}}x_k(\theta) \to J_\theta\bar{x}(\theta)$ as $k \to \infty$. However, it is easy to see that this estimator is exact in the case of exact single-step algorithms, meaning $F$ satisfies $F(x, \theta) = \bar{x}(\theta)$ for all $x, \theta$. Indeed, in this case, one has $J_x F(x, \theta) = 0$ and $J_\theta F(x, \theta) = J_\theta\bar{x}(\theta)$ for all $x, \theta$. Such a situation occurs, for example, when applying Newton's method to an unconstrained quadratic problem. This is a very degenerate situation as it does not really make sense to talk about an "iterative algorithm" in this case. It turns out that this property of being "asymptotically correct" remains valid for very fast algorithms, that is, algorithms that require few iterations to converge, the archetypal example being Newton's method for which we obtain quantitative estimates.

### 2.3.1 Super-linear algorithms

The following is a typical property of fast converging algorithms.

**Assumption 2 (Vanishing Jacobian)** Assume that $F\colon \mathbb{R}^n \times \mathbb{R}^m \to \mathbb{R}^n$ is $C^1$ and that the recursion $x_{k+1} = F_\theta(x_k)$ converges globally, locally uniformly in $\theta$ to the unique fixed point $\bar{x}(\theta)$ of $F_\theta$ such that $J_x F(\bar{x}(\theta), \theta) = 0$.

Note that under Assumption 2, it is always possible to find a small neighborhood of $\bar{x}$ such that $\|J_x F_\theta\|_{\mathrm{op}}$ remains small, that is, Assumption 1 holds locally and Lemma 1 applies. Furthermore, it is possible to show that the derivative estimate is asymptotically correct as follows.

**Corollary 2 (Jacobian convergence)** *Let $F\colon \mathbb{R}^n \times \mathbb{R}^m \to \mathbb{R}^n$ be as in Assumption 2. Then $J^{\mathrm{OS}}x_k(\theta) \to J_\theta\bar{x}(\theta)$ as $k \to \infty$, and $J^{\mathrm{OS}}\bar{x}(\theta) = J_\theta\bar{x}(\theta)$.*

**Proof :** Since $J_x F(\bar{x}(\theta), \theta) = 0$, implicit differentiation of the fixed point equation reduces to $J_\theta\bar{x}(\theta) = J_\theta F(\bar{x}(\theta), \theta)$, and the result follows by continuity of the derivatives. □

**Example 2 (Superlinearly convergent algorithm)** Assume that $F$ is $C^1$ and for each $\rho > 0$, there is $R > 0$ such that $\|F_\theta(x) - \bar{x}(\theta)\| \leq \rho\|x - \bar{x}(\theta)\|$ for all $x, \theta$ such that $\|x - \bar{x}(\theta)\| \leq R$. Then Corollary 2 applies as for any $v$

$$J_x F(\bar{x}(\theta), \theta)v = \lim_{t \to 0} \frac{F(\bar{x}(\theta) + tv, \theta) - \bar{x}(\theta)}{t} = 0.$$

### 2.3.2 Quadratically convergent algorithms

Under additional quantitative assumptions, it is possible to obtain more precise convergence estimates similar to those obtained for implicit differentiation, see Remark 2.

**Corollary 3** *Let $F$ be as in Assumption 2 such that $x \mapsto J_{(x,\theta)}F(x,\theta)$ (joint jacobian in $(x,\theta)$) is $L_J$ Lipschitz (in operator norm). Then, the recursion is asymptotically quadratically convergent and for each $k \geq 1$,*

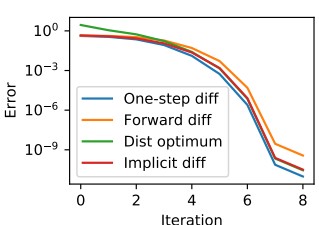

Figure 2: Newton's method quadratic convergence.

$$\|J^{\mathrm{OS}}x_k(\theta) - J_\theta\bar{x}(\theta)\|_{\mathrm{op}} \leq L_J\|x_{k-1}(\theta) - \bar{x}(\theta)\|. \tag{7}$$

**Proof :** Following the same argument as in the proof of Corollary 2, we have

$$\|J^{\mathrm{OS}}x_k(\theta) - J_\theta\bar{x}(\theta)\|_{\mathrm{op}} = \|J_\theta F(x_k(\theta), \theta) - J_\theta F(\bar{x}(\theta), \theta)\|_{\mathrm{op}} \leq L_J\|x_{k-1}(\theta) - \bar{x}(\theta)\|. \tag{8}$$

As for the quadratic convergence, we may assume that $\bar{x}(\theta) = 0$ and drop the $\theta$ variable to simplify notations. We have $F(0) = 0$ and for all $x$,

$$F(x) = \int_0^1 J_x F(tx)x\,dt \leq \|x\| \int_0^1 \|J_x F(tx)\|_{\mathrm{op}}dt \leq \|x\|^2 L_J \int_0^1 t\,dt = \frac{L_J\|x\|^2}{2}.$$

Thus $L_J/2\|x_{k+1}\| \leq [L_J/2\|x_k\|]^2$, and asymptotic quadratic convergence follows. $\square$

**Example 3 (Newton's algorithm)** Assume that $f \colon \mathbb{R}^n \times \mathbb{R}^m \to \mathbb{R}$ is $C^3$ with Lipschitz derivatives, and for each $\theta$, $x \mapsto f(x, \theta)$ is $\mu$-strongly convex. Then Newton's algorithm with backtracking line search satisfies the hypotheses of Corollary 3, see [15, Sec. 9.5.3]. Indeed, it takes unit steps after a finite number of iterations, denoting by $\bar{x}(\theta)$ the unique solution to $\nabla_x f(x, \theta) = 0$, for all $x$ locally around $\bar{x}(\theta)$

$$F(x, \theta) = x - \nabla_{xx}^2 f(x, \theta)^{-1}\nabla_x f(x, \theta).$$

We have, $\nabla_{xx}^2 f(x, \theta)F(x, \theta) = \nabla_{xx}^2 f(x, \theta)x - \nabla_x f(x, \theta)$, differentiating using tensor notations,

$$\nabla_{xx}^2 f(x, \theta)J_x F(x, \theta) = \nabla^3 f(x, \theta)[x, \cdot, \cdot] - \nabla^3 f(x, \theta)[F(x, \theta), \cdot, \cdot]$$

so that $J_x F(\bar{x}(\theta), \theta) = 0$ and Lipschitz continuity of the derivatives of $f$ implies Lipschitz continuity of $J_x F(x, \theta)$ using the fact that $\nabla_{xx}^2 f(x, \theta) \succeq \mu I$ and that matrix inversion is smooth and Lipschitz for such matrices. The quadratic convergence of the three derivative estimates for Newton's method is illustrated on a logistic regression example in Figure 2.

## 3 Hypergradient descent for bilevel problems

Consider the following bilevel optimization problem

$$\min_\theta \quad g(x(\theta)) \quad \text{s.t.} \quad x(\theta) \in \arg\min_y f(y, \theta),$$

where $g$ and $f$ are $C^1$ functions. We will consider bilevel problems such that the inner minimum is uniquely attained and can be described as a fixed point equation $x = F(x, \theta)$ where $F$ is as in Assumption 1. The problem may then be rewritten as

$$\min_\theta \quad g(x(\theta)) \quad \text{s.t.} \quad x(\theta) = F(x(\theta), \theta), \tag{9}$$

| Method | Time | Memory | Error sources |
|---|---|---|---|
| Piggyback recursion | $kn(\omega C_F + nm)$ | $n(n+m)$ | suboptimality + burn-in |
| AD forward-mode | $k\omega C_F m$ | $n+m$ | suboptimality + burn-in |
| AD reverse-mode | $k\omega C_F$ | $kn$ | suboptimality + burn-in |
| Implicit differentiation | $\omega C_F n + n^3$ | $n$ | suboptimality |
| One-step differentiation | $\omega C_F$ | $n$ | suboptimality + lack of contractivity |
| Forward algorithm | $kC_F$ | $n$ | suboptimality |

Table 2: Time and memory complexities of the estimators in Section 2.1 (up to multiplicative constant). $F$ has time complexity denoted by $C_F$ and we consider $k$ iterations in $\mathbb{R}^n$ with $m$ parameter. $\omega$ is the multiplicative overhead of evaluating a gradient (cheap gradient principle).

see illustrations in Remark 1. Gradient descent (or hyper-gradient) on (9) using our one-step estimator in (4) consists in the following recursion

$$\theta_{l+1} = \theta_l - \alpha J^{\mathrm{OS}} x_k(\theta_l)^T \nabla g(x_k(\theta_l)), \qquad (10)$$

where $\alpha > 0$ is a step size parameter. Note that the quantity $J^{\mathrm{OS}} x_k(\theta)^T \nabla g(x_k)$ is exactly what is obtained by applying backpropagation to the composition of $g$ and Algorithm 3, without any further custom variation on backpropagation. Note that one in practice may use amortized algorithms, such as [18]. This section is dedicated to the theoretical guarantees which can be obtained using such a procedure, proofs are postponed to Appendix A.

### 3.1 Complexity analysis of different hypergradient strategies

We essentially follow the complexity considerations in [27, Section 4.6]. Let $C_F$ denote the computation time cost of evaluating the fixed-point map $F$ and $\omega > 0$ be the multiplicative overhead of gradient evaluation, in typical applications, $\omega \leq 5$ (cheap gradient principle [6]). The time cost of evaluating the Jacobian of $F$ is $n\omega C_F$ ($n$ gradients). Forward algorithm evaluation (i.e., $x_k$) has computational time cost $kC_F$ with a fixed memory complexity $n$. Vanilla piggyback recursion (2) requires $k-1$ full Jacobians and matrix multiplications of costs $n^2 m$. The forward-mode of AD has time complexity $k\omega C_F m$ (compute $m$ partial derivatives each of them cost $\omega$ times the time to evaluate the forward algorithm), and requires to store the iterate vector of size $n$ and $m$ partial derivatives. The reverse-mode of AD has time complexity $k\omega C_F$ (cheap gradient principle on $F_\theta^k$) and requires to store $k$ vectors of size $n$. Implicit differentiation requires *one* full Jacobian ($\omega C_F n$) and solution of *one* linear system of size $n \times n$, that is roughly $n^3$. Finally, one-step differentiation is given by only differentiating a single step of the algorithm at cost $\omega C_F$. For each estimate, distance to the solution will result in derivative errors. In addition, automatic differentiation based estimates may suffer from the burn-in effect [41] while one-step differentiation will suffer from a lack of contractivity as in Corollary 1. We summarize the discussion in Table 2. Let us remark that, if $C_F \geq n^2$, then reverse AD has a computational advantage if $k \leq n$, which makes sense for fast converging algorithms, but in this case, one-step differentiation has a small error and a computational advantage compared to reverse AD.

**Remark 3 (Implicit differentiation: but on which equation?)** Table 2 is informative yet formal. In practical scenarios, implicit differentiation should be performed using the simplest equation available, not necessarily $F = 0$. This can significantly affect the computational time required. For instance, when using Newton's method $F = -[\nabla^2 f]^{-1} \nabla f$, implicit differentiation should be applied to the gradient mapping $\nabla f = 0$, not $F$. In typical application $C_{\nabla f} = O(n^2)$, and the dominant cost of implicit differentiation is $O(n^3)$, which is of the same order as the one-step differentiation as $C_F = O(n^3)$ (a linear system needs to be inverted). However, if the implicit step was performed on $F$ instead of $\nabla f$, it would incur a prohibitive cost of $O(n^4)$. In conclusion, the implicit differentiation phase is not only custom in terms of the implementation, but also in the very choice of the equation.

## 3.2 Approximation analysis of one step differentiation

The following corollary, close to [42, Prop. 3.1], provides a bound on the one-step differentiation (Algorithm 3) gradient estimator for (9). The bound depends on $\rho$, the contraction factor, and distance to the solution for the inner algorithm. The proof is given in Appendix A.

**Corollary 4** *Let $F \colon \mathbb{R}^n \times \mathbb{R}^m \to \mathbb{R}^n$ be as in Corollary 1 and consider the bilevel problem (9), where $g$ is a $C^1$, $l_g$ Lipschitz function with $l_\nabla$ Lipschitz gradient. Then,*

$$\left\| \nabla_\theta (g \circ \bar{x})(\theta) - J^{\mathrm{OS}} x_k(\theta)^T \nabla g(x_k) \right\| \leq \frac{\rho L_F l_g}{1 - \rho} + L_J l_g \|x_{k-1} - \bar{x}(\theta)\| + L_F l_\nabla \|\bar{x}(\theta) - x_k\|.$$

## 3.3 Approximate critical points

The following lemma is known, but we provide a proof for completeness in Appendix A.

**Lemma 3** *Assume that $h \colon \mathbb{R}^m \to \mathbb{R}$ is $C^1$ with $L$ Lipschitz gradient and lower bounded by $h^*$. Assume that for some $\epsilon > 0$, for all $l \in \mathbb{N}$, $\left\| \theta_{l+1} - \theta_l + \frac{1}{L} \nabla h(\theta_l) \right\| \leq \frac{\epsilon}{L}$. Then, for all $K \in \mathbb{N}$, $K \geq 1$, we have*

$$\min_{l=0,\ldots,K} \|\nabla h(\theta_l)\|^2 \leq \epsilon^2 + \frac{2L(h(x_0) - h^*)}{K + 1}.$$

Combining with Corollary 4, it provides a complexity estimate for the recursion in (10).

**Corollary 5** *Under the setting of Corollary 4, consider iterates in (10) with $k \geq 2$ and assume the following*

- $\sup_\theta \|x_0(\theta) - F_\theta(x_0(\theta))\| \leq M$, *for some $M > 0$.*
- $F$ *is $L_F$ Lipschitz and $J_{(x,\theta)} F$ is $L_J$ Lipschitz jointly in operator norm.*
- $g$ *is $C^1$, $l_g$ Lipschitz with $l_\nabla$ Lipschitz gradient.*
- $\frac{1}{\alpha} \geq \left( \frac{L_J}{1-\rho} \left( \frac{L_F}{1-\rho} + 1 \right) l_g + l_\nabla \frac{L_F}{1-\rho} \right) \frac{L_F}{1-\rho}$
- $g \circ \bar{x}$ *is lower bounded by $g^*$.*

*Then setting $\epsilon = \frac{\rho}{1-\rho}(L_F l_g + (L_J l_g + L_F l_\nabla) M \rho^{k-2})$, for all $K \in \mathbb{N}$,*

$$\min_{l=0,\ldots,K} \|\nabla_\theta (g \circ \bar{x})(\theta_l)\|^2 \leq \epsilon^2 + \frac{2L((g \circ \bar{x})(\theta_0) - g^*)}{K + 1}.$$

The level of approximate criticality is the sum of a term proportional to $\rho$ and a term inversely proportional to $K$. For large values of $k$ (many steps on the inner problem), $K$ (many steps on the outer problem), approximate criticality is essentially proportional to $\rho L_F l_g / (1 - \rho)$ which is small if $\rho$ is close to 0 (*e.g.* superlinear algorithms).

# 4 Numerical experiments

We illustrate our findings on three different problems. First, we consider Newton's method applied to regularized logistic regression, as well as interior point solver for quadratic problems. These are two fast converging algorithms for which the results of Section 2.3 can be applied and the one-step procedure provides accurate estimations of the derivative with a computational overhead negligible with respect to solution evaluation, as for implicit differentiation. We then consider the gradient descent algorithm applied to a ridge regression problem to illustrate the behavior of the one step procedure for linearly convergent algorithms.

**Logistic regression using Newton's algorithm.** Let $A \in \mathbb{R}^{N \times n}$ be a design matrix, the first column being made of 1s to model an intercept. Rows are denoted by $(a_1, \ldots, a_N)$. Let $x \in \mathbb{R}^n$ and $y \in \{-1, 1\}^N$. We consider the regularized logistic regression problem

$$\min_{x \in \mathbb{R}^n} \sum_{i=1}^N \theta_i \ell(\langle a_i, x \rangle \, y_i) + \lambda \|x_{-1}\|^2, \tag{11}$$

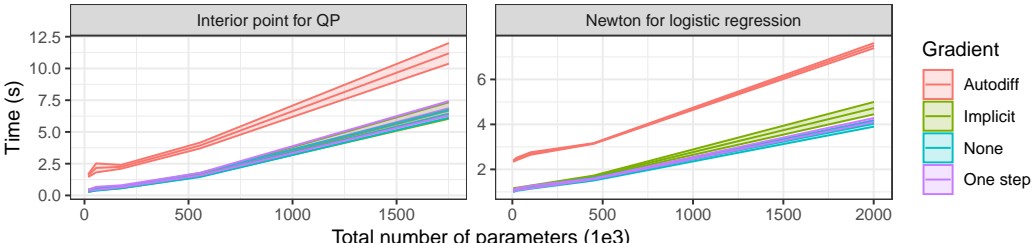

Figure 3: Timing experiment to evaluate one gradient. Left: Differentiable QP in (12), one step and implicit estimators agree up to an error of order $10^{-16}$. Right: Newton's method on logistic regression (11), one step and implicit estimators agree up to an error of order $10^{-12}$. Label "None" represent solving time and "Autodiff", "Implicit" and "One step" represent solving time and gradient evaluation for each estimator in Section 2.1. The mean is depicted using shading to indicate standard deviation estimated over 10 runs.

where $\ell$ is the logistic loss, $\ell \colon t \mapsto \log(1 + \exp(-t))$, $\lambda > 0$ is a regularization parameter, and $x_{-1}$ denotes the vector made of entries of $x$ except the first coordinate (we do not penalize intercept). This problem can be solved using Newton's method which we implement in `jax` using backtracking line search (Wolfe condition). Gradient and Hessian are evaluated using `jax` automatic differentiation, and the matrix inversion operations are performed with an explicit call to a linear system solver.

We denote by $x(\theta)$ the solution to problem (11) and try to evaluate the gradient of $\theta \mapsto \|x(\theta)\|^2 / 2$ using the three algorithms presented in Section 2.1. We simulate data with Gaussian class conditional distributions for different values of $N$ and $n$. The results are presented in Figure 3 where we represent the time required by algorithms as a function of the number of parameters required to specify problem (11), in our case size of $A$ and size of $y$, which is $(n + 1)N$.

Figure 3 illustrates that both one-step and implicit differentiation enjoy a marginal computational overhead, contrary to algorithmic differentiation. In this experiment, the one-step estimator actually has a slight advantage in terms of computation time compared to implicit differentiation.

**Interior point solver for quadratic programming:** The content of this section is motivated by elements described in [4], which is associated with a `pytorch` library implementing a standard interior point solver. Consider the following quadratic program (QP):

$$\min_{x \in \mathbb{R}^n} \quad \frac{1}{2} x^T Q x + q^T x \quad \text{s.t.} \quad Ax = \theta, \quad Gx \leq h, \tag{12}$$

where $Q \in \mathbb{R}^{n \times n}$ is positive definite, $A \in \mathbb{R}^{m \times n}$ and $G \in \mathbb{R}^{p \times n}$ are matrices, $q \in \mathbb{R}^n$, $\theta \in \mathbb{R}^m$ and $h \in \mathbb{R}^p$ are vectors. We consider $x(\theta)$ the solution of problem (12) as a function of $\theta$, the right-hand side of the equality constraint. We implemented in `jax` a standard primal-dual Interior Point solver for problem (12). Following [4], we use the implementation described in [43], and we solve linear systems with explicit calls to a dedicated solver. For generic inputs, this algorithm converges very fast, which we observed empirically. Differentiable programming capacities of `jax` can readily be used to implement the automatic differentiation and one-step derivative estimators without requiring custom interfaces as in [4]. Indeed, implicit differentiation for problem (12) was proposed in [4] with an efficient `pytorch` implementation. We implemented these elements in `jax` in order to evaluate $J_\theta x(\theta)$ using implicit differentiation. More details on this experiment are given in Appendix B.

We consider evaluating the gradient of the function $\theta \mapsto \|x(\theta)\|^2 / 2$ using the three algorithms proposed in Section 2.1. We generate random instances of QP in (12) of various sizes. The number of parameters needed to describe each instance is $n(n + 1) + (n + 1)m + (n + 1)p$. The results are presented in Figure 3 where we represent the time required by algorithms as a function of the number of parameters required to specify problem (12). In all our experiments, the implicit and one-step estimates agree up to order $10^{-6}$. From Figure 3, we

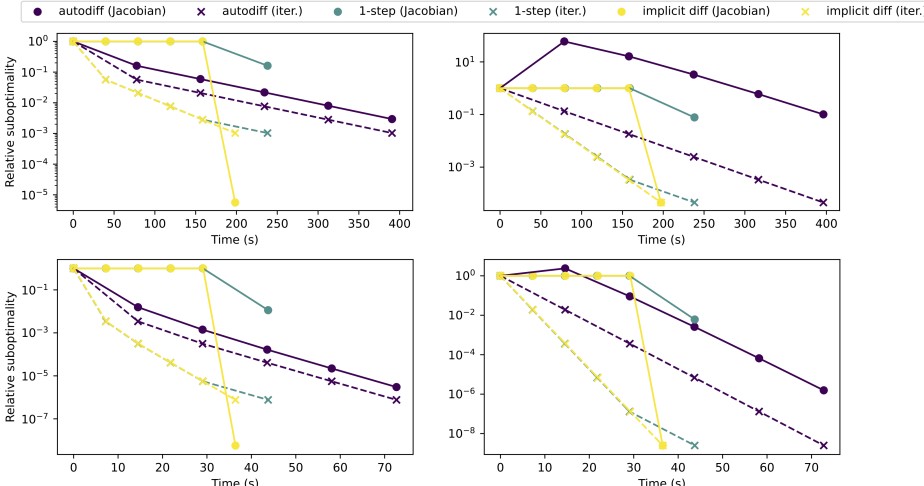

Figure 4: Differentiation of gradient descent for solving weighted Ridge regression on `cpusmall`. Top line: condition number of 1000. Bottom line: condition number of 100. Left column: small learning rate $\frac{1}{L}$. Right column: big learning rate $\frac{2}{\mu+L}$. Dotted (resp. filled) lines represent the error of the iterates (resp. of the Jacobians).

see that both one-step and implicit differentiation enjoy a marginal additional computational overhead, contrary to algorithmic differentiation.

**Weighted ridge using gradient descent.** We consider a weighted ridge problem with $A \in \mathbb{R}^{N \times n}$, $y \in \mathbb{R}^N$, $\lambda > 0$ and a vector of weights $\theta \in \mathbb{R}^N$:

$$\bar{x}(\theta) = \arg \min_{x \in \mathbb{R}^n} f_\theta(x) = \frac{1}{2} \sum_{i=1}^N \theta_i (y_i - \langle a_i, x \rangle)^2 + \frac{\lambda}{2} \|x\|^2.$$

We solve this problem using gradient descent with adequate step-size $F(x, \theta) = x - \alpha \nabla f_\theta(x)$ with $x_0(\theta) = 0$, and we consider the $K$-step truncated Jacobian propagation $\tilde{F} = F_\theta^K$ with $K = 1/\kappa$ where $\kappa$ is the effective condition number of the Hessian. Figure 4 benchmarks the use of automatic differentiation, one-step differentiation, and implicit differentiation on the data set `cpusmall` provided by LibSVM [17] for two types of step-sizes. We monitor both quantities $\|x_k(\theta) - \bar{x}(\theta)\|^2$ for the iterates, and $\|J_\theta x_k(\theta) - J_\theta \bar{x}(\theta)\|^2$ for the Jacobian matrices. As expected, implicit differentiation is faster and more precise, it is our gold standard which requires *custom implicit system to be implemented* (or the use of an external library). For large steps, autodiff suffers from the burn-in phenomenon described in [41], which does not impact the one step estimator. Therefore, for a fixed time budget, the one step strategies allows to obtain higher iterate accuracy and similar, or better, Jacobian accuracy. In the small step regime, one step differentiation provides a trade-off, for a fixed time budget, one obtains better estimates for the iterate and worse estimates for the Jacobian matrices. Our results suggest that $K$-step truncated backpropagation allows to save computation time, at the cost of possibly degraded derivatives compared to full backpropagation, in line with [42].

## 5 Conclusion

We studied the one-step differentiation, also known as Jacobian-free backpropagation, of a generic iterative algorithm, and provided convergence guarantees depending on the initial rate of the algorithm. In particular, we show that one-step differentiation of a quadratically convergent algorithm, such as Newton's method, leads to a quadratic estimation of the Jacobian. A future direction of research would be to understand how to extend our findings to the nonsmooth world as in [14] for linearly convergent algorithms.

## Acknowledgements

The authors acknowledge the support of the AI Interdisciplinary Institute ANITI funding, through the French "Investments for the Future – PIA3" program under the grant agreement ANR-19-PI3A0004, Air Force Office of Scientific Research, Air Force Material Command, USAF, under grant numbers FA8655-22-1-7012, ANR MaSDOL 19-CE23-0017-0, ANR Chess, grant ANR-17-EURE-0010, ANR Regulia, ANR GraVa ANR-18-CE40-0005. Jérôme Bolte, Centre Lagrange, and TSE-P.

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

# A    Proof of Section 3

**Proof of Corollary 4:**  We have $\nabla_\theta (g \circ \bar{x})(\theta) = J_\theta \bar{x}(\theta)^T \nabla g(\bar{x}(\theta))$, and

$$J_\theta \bar{x}(\theta)^T \nabla g(\bar{x}(\theta)) - J^{\mathrm{OS}} x_k(\theta)^T \nabla g(x_k) = (J_\theta \bar{x}(\theta)^T - J^{\mathrm{OS}} x_k(\theta)^T) \nabla g(\bar{x}(\theta))$$
$$- J^{\mathrm{OS}} x_k(\theta)^T (\nabla g(x_k) - \nabla g(\bar{x}(\theta))).$$

The result follows from the triangular inequality combined with Corollary 1 and the following

$$\|(J_\theta \bar{x}(\theta)^T - J^{\mathrm{OS}} x_k(\theta)^T) \nabla g(\bar{x}(\theta))\| \leq \|J_\theta \bar{x}(\theta) - J^{\mathrm{OS}} x_k(\theta)\|_{\mathrm{op}} \|\nabla g(\bar{x}(\theta))\|$$
$$\leq l_g \|J_\theta \bar{x}(\theta) - J^{\mathrm{OS}} x_k(\theta)\|_{\mathrm{op}},$$

and

$$\|J^{\mathrm{OS}} x_k(\theta)^T (\nabla g(x_k) - \nabla g(\bar{x}(\theta))\| \leq \|J^{\mathrm{OS}} x_k(\theta)\|_{\mathrm{op}} \|\nabla g(x_k) - \nabla g(\bar{x}(\theta))\| \leq L_F l_\nabla \|\bar{x}(\theta) - x_k\|$$

$\square$

**Proof of Lemma 3:**  We have, using the "descent lemma" (see [7]), for all $l \in 0 \ldots K$,

$$h(\theta_{l+1}) - h(\theta_l) \leq \langle \nabla h(\theta_l), \theta_{l+1} - \theta_l \rangle + \frac{L}{2} \|\theta_{l+1} - \theta_l\|^2$$
$$= \frac{L}{2} \left( 2 \left\langle \frac{\nabla h(\theta_l)}{L}, \theta_{l+1} - \theta_l \right\rangle + \|\theta_{l+1} - \theta_l\|^2 \right)$$
$$= \frac{L}{2} \left( \left\| \theta_{l+1} - \theta_l + \frac{\nabla h(\theta_l)}{L} \right\|^2 - \left\| \frac{\nabla h(\theta_l)}{L} \right\|^2 \right)$$
$$\leq \frac{1}{2L} \left( \epsilon^2 - \min_{l=0,\ldots,K} \|\nabla h(\theta_l)\|^2 \right).$$

Summing for $l = 0, \ldots, K$, we have

$$h^* - h(\theta_0) \leq h(\theta_{K+1}) - h(\theta_0) \leq \frac{K+1}{2L} \left( \epsilon^2 - \min_{l=0,\ldots,K} \|\nabla h(\theta_l)\|^2 \right).$$

and we deduce, by using concavity of the square root, that

$$\min_{l=0,\ldots,K} \|\nabla h(\theta_l)\|^2 \leq \epsilon^2 + \frac{2L(h(x_0) - h^*)}{K+1}.$$

$\square$

**Lemma 4** *Let $A \colon \mathbb{R}^d \to \mathbb{R}^{n \times m}$ and $B \colon \mathbb{R}^d \to \mathbb{R}^{m \times p}$, be $M_A$ and $M_B$ bounded and $L_A$ and $L_B$ Lipschitz respectively, in operator norm, then $A \cdot B$ is $M_A L_B + L_A M_B$ Lipschitz in operator norm.*

**Proof :** We have for all $x, y \in \mathbb{R}^d$

$$\|A(x)B(x) - A(y)B(y)\|_{\mathrm{op}} = \|(A(x) - A(y))B(x) - A(y)(B(y) - B(x))\|_{\mathrm{op}}$$
$$\leq \|(A(x) - A(y))B(x)\|_{\mathrm{op}} + \|A(y)(B(y) - B(x))\|_{\mathrm{op}}$$
$$\leq \|A(x) - A(y)\|_{\mathrm{op}} \cdot \|B(x)\|_{\mathrm{op}} + \|A(y)\|_{\mathrm{op}} \cdot \|B(y) - B(x)\|_{\mathrm{op}}$$
$$\leq (L_A M_B + M_A L_B) \|x - y\|$$

$\square$

**Lemma 5** *The function $M \mapsto (I - M)^{-1}$ is $(1-\rho)^{-2}$ Lipschitz in operator norm on the set of square matrices $M$ such that $\|M\|_{\mathrm{op}} \leq \rho < 1$.*

**Proof :** For any two square matrices $A, B$ with operator norm bounded by $\rho < 1$, $I - A$ and $I - B$ are invertible and the operator norm of the inverse is at most $(1-\rho)^{-1}$ (see proof of

Lemma 2). We have

$$(I - A) \left( (I - A)^{-1} - (I - B)^{-1} \right) (I - B) = A - B$$
$$(I - A)^{-1} - (I - B)^{-1} = (I - A)^{-1}(A - B)(I - B)^{-1}$$
$$\|(I - A)^{-1} - (I - B)^{-1}\|_{\mathrm{op}} = \|(I - A)^{-1}(A - B)(I - B)^{-1}\|_{\mathrm{op}}$$
$$\leq \|(I - A)^{-1}\|_{\mathrm{op}} \cdot \|(A - B)\|_{\mathrm{op}} \cdot \|(I - B)^{-1}\|_{\mathrm{op}}$$
$$\leq \frac{\|(A - B)\|_{\mathrm{op}}}{(1 - \rho)^2}$$

$\square$

**Proof of Corollary 5:** Let us evaluate a Lipschitz constant of $\nabla_\theta(g \circ \bar{x})$. We have

$$\nabla_\theta(g \circ \bar{x})(\theta) = J_\theta \bar{x}(\theta)^T \nabla g(\bar{x}(\theta))$$
$$= \left( I - J_x F(\bar{x}, \theta) \right)^{-1} J_\theta F(\bar{x}, \theta) \right)^T \nabla g(\bar{x}(\theta)).$$

This can be seen as a composition of a matrix function with $\bar{x}$. Combining Lemma 4 and Lemma 5 with the fact that $J_x F$ and $J_\theta F$ are both $L_J$ Lipschitz and $L_F$ bounded, the function $(x, \theta) \mapsto (I - J_x F(x, \theta))^{-1} J_\theta f(x, \theta)$ is $\frac{L_J L_F}{(1-\rho)^{-2}} + \frac{L_J}{1-\rho} = \frac{L_J}{1-\rho} \left( \frac{L_F}{1-\rho} + 1 \right)$ Lipschitz.

Invoking Lemma 4 and using the fact that the operator norm of a vector is the euclidean norm, we have that the function $(x, \theta) \mapsto (I - J_x F(x, \theta))^{-1} J_\theta f(x, \theta) \nabla g(x)$ is $\frac{L_J}{1-\rho} \left( \frac{L_F}{1-\rho} + 1 \right) l_g + l_\nabla \frac{L_F}{1-\rho}$ Lipschitz.

From the implicit function theorem, $\|J_\theta \bar{x}(\theta)\|_{\mathrm{op}} \leq L_F/(1 - \rho)$ so that $\bar{x}$ is $L_F/(1 - \rho)$ Lipschitz and $\nabla_\theta(g \circ \bar{x})$ is $\left( \frac{L_J}{1-\rho} \left( \frac{L_F}{1-\rho} + 1 \right) l_g + l_\nabla \frac{L_F}{1-\rho} \right) \frac{L_F}{1-\rho}$ Lipschitz so the choice of $\alpha$ is such that $1/\alpha$ a Lipschitz constant for the gradient.

Using Corollary 4, we are in the setting of Lemma 3 and the result follows. $\square$

## B  Experimental details Section 4

### B.1  QP experiment

**Implicit differentiation:** We explicitly formed the Jacobians of the iterative process (function $f$) and solved the underlying linear systems using a solver in `jax`. The inversion was not done based on VJP (using fixed point iterations or conjugate gradient) because, for the relatively small scale of our problems, pure linear algebra was more efficient. We did not form the full implicit Jacobian in equation (3) using full matrix inversion; we only solved a linear system involving a left incoming vector in equation (3). In this sense, we did implement VJP for implicit differentiation although we explicitly formed Jacobians because this was the most efficient.

**Generation of random QPs:** To generate the QP instances, the quantities $n, m, p$ are varied on a logarithmic scale. We set $Q = M^T M$ where $M$ is of size $n \times n$ with entries uniform in $[-1, 1]$. Constraint matrices $A$ and $G$ also have entries uniform in $[-1, 1]$, and we chose $m$ and $p$ to be smaller than $n$ so that feasibility is generic and occurs with probability one.

