# OpenReview forum: "One-step differentiation of iterative algorithms"
_NeurIPS.cc/2023/Conference — NeurIPS 2023 spotlight_

### Official Review · Reviewer_qFET · 2023-06-29

**Soundness:** 3 good
**Presentation:** 3 good
**Contribution:** 2 fair
**Rating:** 7
**Confidence:** 5

**Summary:**

In bilevel optimization, or optimization problems with equilibrium constraints, computing a derivative of the upper-level problem is a well-known stumbling block as this requires "differentiating through" the solution to the lower level problem.

This paper provides a theoretical study for one approach to overcoming this issue, __one-step differentiation__, also known as Jacobian-free Backpropagation. Although one-step differentiation is deceptively simple, the core contribution of this paper is that it is safe to use in many situations.

Theoretical results are bolstered by some numerical results. In addition there are several nice examples, remarks, and corollaries that could be of use to practitioners (for example, how to improve the quality of the one-step derivative by using a k-step approach instead).

**Strengths:**

 - In my opinion, this paper's biggest strength is that it has a simple message ("One-step differentiation works in many situations") and conveys this message in a clear and accessible to non-experts way.
  - There are several little gems of wisdom sprinkled throughout this paper, e.g. applying one-step differentiation to $F^K$ instead of $F$ (top of pg. 4), using different operators for the forward and backward pass for implicit differentation (Remark 2).
 - I like the bound on hypergradient approximation (Corollary 4).
 - The clear comparisons between the complexities of various gradient estimation approaches contained in Table 2 is great.

**Weaknesses:**


1. In addition to Automatic Differentiation (AD) and Implicit Differentiation (ID), there is a third benchmark approach the authors should consider, namely _Inexact Automatic Differentiation_ (IAD) (see for example [1,2]). In particular, [2] makes a surprising connection between IAD and ID, showing they are essentially the same. I'd like to see some discussion of the IAD approach in this paper, and perhaps even an inclusion of this approach into the numerical experiments.
2. See "Questions" section below for some questions on the relationship between your work and that of [3].

*Minor Issues*

3. the phrase "piggyback recursion" is used for the first time in line 207. I suggest mentioning this terminology immediately after equation (2) and providing a reference.
4. In addition to [34] (Vlastelica et al), I'd suggest citing [4], which predates [34] and also considers one-step differentiation for polytope-constrained lower-level problems.
5. The notation in Lemma 3 is confusing. I'd strongly suggest using $\theta_i$ instead of $x_k$ and $g$ instead of $f$, as the Lemma is intended to be applied to the upper level problem.
6. I have several comments regarding Fig. 3:
      - The time displayed is to calculate a single gradient, right? If yes, this should be made clearer in the caption.
      - The x-axis needs to be changed. I'd suggest $5\times 10^{5}$ instead of 50000 and so on.
      - What does the shading represent?
      - See comment 1 about Inexact Automatic Differentiation.
7. Could you say more about how you randomly generate QP instances (pg 8--9) in particular, how do you ensure feasibility?

*Typos etc.*

8. "well-foundness" in line 10 should be "well-foundedness".
9. "a important" in line 12 should be "an important".
10. "relies" on line 20 should be "rely".
11. "distance with" on lone 33 should be "distance to".
12. "quantiative" on line 58 should be "quantitative"
13. "parameteric" on line 72 should be "parametric"
14. "Superlinar" on line 144 should be "Superlinear"
15. "guaranties" on line 199 should be "guarantees"

[1] _Automatic Differentiation of Some First-Order Methods in Parametric Optimization_ by Mehmood and Ochs (2019)

[2] _Analyzing Inexact Hypergradients for Bilevel Learning_ by Ehrhardt and Roberts (2023)

[3] _JFB: Jacobian-Free Backpropagation for Implicit Networks_ by Wu Fung et al (2022)

[4] _Learn to Predict Equilibria via Fixed Point Networks_ by Heaton et al (2021)


**Questions:**

In [3] implicit networks with a head and tail ($S_{Theta}$ and $Q_{Theta}$ in their notation) are considered. This complicates the analysis of the hypergradient, see the proof of Theorem 3.1, particularly when analyzing the inner product between the true hypergradient and the approximation $p_{\Theta}$.

 Your analysis of hypergradients (i.e. Corollary 4) works out nicely, partly because you consider the distance between true and approx. hypergradient, not the inner product. Can you apply this to get quantitative bounds for the setting considered in [3]?


[3] _JFB: Jacobian-Free Backpropagation for Implicit Networks_ by Wu Fung et al (2022)

**Limitations:**

Yes.

---

> ### Author Rebuttal · Authors · 2023-08-04
>
> We greatly thank the referee for his detailed comments and evaluation of our manuscript.
>
> 1. We will mention Inexact AD and discuss the references proposed by the referee in the related work section. In our situation, it is not really clear how to apply inexact AD. Indeed, inexact AD is often applied to first-order methods, where the evaluation of the Jacobian of a single iteration is cheap, and many iterations are performed. In the context of Figure 3, we consider super-linearly convergent algorithms, typically variations of Newton's method. In this setting, evaluation of a single iteration is costly (typically similar as implicit differentiation) with very few iterations performed. One step differentiation is "exact" in the sense that it finds the same derivative as implicit differentiation up to numerical errors (in our experiments) while inexact AD will necessarily have some time overhead as it requires more operations compared to one-step differentiation (form the full Jacobian and perform fixed point iterations). We, therefore, believe that the settings where one-step differentiation has good performances are not favorable settings for inexact AD (and vice versa). We see the two approaches as complementary, which we will discuss in a revised version of the paper. For this reason, we will not include timing experiments for inexact AD as the comparison would not be really fair (for the same reason, we did not include forward AD, or unrolling, in our experiments).
>
> 3. Indeed, we will mention the term piggyback right after (2).
> 4. We will include the cited references in the related work section.
> 5. The referee is right, Lemma 3 will be modified, and we will add a corollary to specify it for bilevel problems.
> 6. We will take the remarks of the referee into account regarding Figure 3. In particular, the shaded area is the standard deviation over 10 repetitions of the experiment.
> 7. To generate the QP instances, the quantities $n, m, p$ are varied on a logarithmic scale. We set $Q = M^TM$ where $M$ is of size $n \times n$ with entries uniform in [-1,1]. Constraint matrices $A$ and $G$ also have entries uniform in [-1,1], and we chose $m$ and $p$ to be smaller than $n$ so that feasibility is generic and occurs with probability one. This will be discussed in a dedicated appendix.
> We will correct all the typos pointed out by the referee.
>
> Question:
> Thanks for this nice question. The ideas developed in [3] are indeed similar, but the theoretical results are quite different. The main assumption in [3] is (3.3), which is pretty different from ours:
> - It does not put any constraint on the magnitude of the derivative with respect to parameters (theta).
> - It requires good conditioning: Jacobian matrices (with respect to parameters theta) should be close to identity, and the level of closedness should be balanced with respect to the contraction factor ($\gamma$ in [3]).
>
> The results are, therefore, quite different in nature (descent direction versus quantitative estimation). We will not be able to obtain quantitative estimation under the exact same setting as [3] (since assumptions are different), but we will be able to obtain quantitative estimates under suitable assumptions. These will be exactly the same as in [3] and will be essentially the same as what we developed in Corollary 4. We will add a remark regarding this after Corollary 4 and in comparison with the existing work section.

---

> > ### Comment · Reviewer_qFET · 2023-08-13
> >
> > Great, thanks for addressing all my questions! I have no further comments and hope to see this paper accepted.

---

### Official Review · Reviewer_agCr · 2023-07-06

**Soundness:** 3 good
**Presentation:** 4 excellent
**Contribution:** 3 good
**Rating:** 6
**Confidence:** 4

**Summary:**

In this paper, the authors consider the problem of differentiating the fixed-point of an algorithm with recursive update, with respect to some parameter of the algorithm. They propose a one-step automatic differentiation technique where, once having approximated the fixed-point through the recursive algorithm, they back-propagate only through the last step. They provide a theoretical and numerical comparison of their one-step technique with the existing Automatic and Implicit Differentiation techniques. They show that the technique converges for superlinear algorithms, and incurs an error for linear algorithms which depends on the rate of convergence. For linear algorithms, they propose to use K-step technique where K depends on the condition number. They compare the time complexity of the three strategies on the Newton's method applied to weighted logistic regression and the interior point method applied to constrained quadratic programming problem. They also show the K-step technique on gradient descent applied to weighted ridge regression.

**Strengths:**

-> For super linearly convergent algorithms and for fast linear algorithms, the authors demonstrate
	* Easy implementation of Automatic Differentiation, and
	* Time and Memory complexity of Implicit differentiation.

-> The derivative error of their one-step technique is shown to be depended on the error in the estimation of the solution and the convergence rate (which is zero for superlinear algorithms).

-> Except for the third example in Section 4 and in particular Figure~4, the paper is well-written and structured and very easy to follow. The proofs are quite easy to understand.

**Weaknesses:**

-> For large scale applications (n >> 1), memory and time complexities of one update step of super-linear algorithms do not scale well with n:

- Time: O(n^2) for quasi-newton methods, O(n^3) for Newton methods.
- Memory: O(n^2) for both.

-> In such cases, we resort to the first order methods where the convergence rate is non-zero. For ill-conditioned problems, the authors suggest K-step back-propagation with K = 1/\kappa but then that is just truncated back-propagation and we still have a memory overhead.


**Questions:**

-> In implementations of Algorithms 1, 2 and 3 for Bilevel Optimization, did the authors first construct the jacobians and then evaluate J^T \grad g? Because that is quite inefficient. For bilevel Optimization, implementing VJP for each algorithm is the suitable way. The authors should mention that.

-> In Table 2, I think the authors should clarify that the time complexity of Piggyback recursion (2) is not the same as that of the forward mode AD recursion (1). I believe that is kn\omega C_F. Similarly, the memory complexity is of forward mode AD is n.

-> Figure 4 lacks explaination. I understand that the weighted ridge regression example in Section 4 is aimed to depict the behaviour of the three techniques on gradient descent applied to an ill-conditioned problem. But I don't really understand what is shown in Figure 4.

**Limitations:**

Not applicable.

---

> ### Author Rebuttal · Authors · 2023-08-04
>
> We thank the referee for his thoughtful comments.
>
> Indeed, super linearly convergent algorithms / second-order like methods are fast, but the price to pay is that each step is very expensive (compared to first-order methods, for example). There is no free lunch, and in that case, our analysis only makes sense practically for relatively small-scale scenarios or very well-conditioned problems, but this is a problem-dependent feature, and one does not really have control of this.
> Regarding the memory overhead of K step differentiation, there is indeed an overhead, but it could be relatively small if K remains not too large (e.g., well-conditioned problems).
>
> Questions:
> - We did form Jacobians of the iterative process (F) and solved linear systems using a dedicated solver (in Jax). The inversion was not done based on VJP (using fixed point iterations or conjugate gradient) because, for the relatively small scale of our problems, linear algebra was more efficient.
> - We did not form the full implicit Jacobian in equation (3) using full matrix inversion; we only solved a linear system involving a left incoming vector in equation (3). In this sense, we did implement VJP for implicit differentiation.
> - We will make this precise in a dedicated appendix in the revised version.
>
> Point 2: The reviewer is right; we will add a line with an evaluation of forward AD, which should be slightly more favorable than the piggyback recursion as Jacobian matrices are not explicitly formed and multiplied.
>
> Point 3: We agree and will revise the discussion in the weighted ridge regression section and better explain the content of Figure 4 with more detailed comments.

---

### Official Review · Reviewer_USU8 · 2023-07-06

**Soundness:** 3 good
**Presentation:** 3 good
**Contribution:** 3 good
**Rating:** 7
**Confidence:** 3

**Summary:**

The paper presents a method called one-step differentiation, also known as Jacobian-free backpropagation, as an alternative to automatic and implicit differentiation. The authors analyze the theoretical approximation of one-step Jacobian and provide specific examples such as Newton's method and gradient descent, on which the authors showed explicit convergence of the one-step case to ground truth solution.. They demonstrate the efficiency of one-step differentiation in bilevel optimization and provide numerical illustrations using logistic regression, interior point solver for quadratic programming, and weighted ridge regression with gradient descent. On these examples, the authors demonstrate the efficiency of the one step approach while demonstrating clear speed advantage over the autodiff approach.

**Strengths:**

Automatic differentiation of iterative system is long known for its slowness, rendering many differentiable systems useless in practice. Of course, one can re-route to implicit differentiation, also known as sensitivity analysis, adjoint method, etc. However, implementing them is not fun, especially for complex systems. The authors proposed a competitive alternative that is both easy to implement, fast to compute (figure 3), and, without any loss for accuracy (line 284). I can see it laying the foundation for many downstream applications.

Like author pointed out, using just the jacobian of the last step seems like a too naive approximation. However, the author proved several crucial theorems in the paper showing that it's in fact quite accurate in many cases (of course not all the cases).

The result is general in the sense that it applies to all iterative methods that have a fixed point.

The paper is well written, with a good balance of text, code, and proofs, making the paper easy to follow. That said, I admit that I did not check every step of the proof carefully.

**Weaknesses:**

As a theoretical paper, I really don't see much weakness. That said, I would be interested in seeing validation and experiments on more complex engineering system, potentially involving highly non-convex neural networks.

Minor:
Figure 1 caption, "is explicited". explicit is not a verb.

**Questions:**

line 134 can the author please discuss more when the practioners are advised to use the K last step? in other words, K-step differentiation of iterative algorithm.

**Limitations:**

The paper lacks a discussion of potential limitations or scenarios where the one-step differentiation method may not be suitable or may exhibit suboptimal performance.

---

> ### Author Rebuttal · Authors · 2023-08-04
>
> We thank the referee for his positive feedback. Experiments and validation on highly nonconvex neural network is something that we are currently investigating. From our intuition so far (and as predicted by the theory we developed), one-step differentiation is not a panacea that will perfectly work under all circumstances, but it may have some important benefits for specific architectures under specific settings. We do not have solid enough findings in this direction to include in the present work, but we will for sure invest efforts in this direction.
>
> Of course, we will correct the typo and include a discussion regarding situations for which using K-last steps differentiation is advisable. This will also provide an answer to the reviewer's last: the typical scenario for which one-step differentiation may fail is for slow algorithms, which corresponds to a contraction factor close to 1 (or even equal to 1, in this case, even implicit differentiation has no guarantee). This is actually the reason why we propose to differentiate K steps instead of a single step in this situation. This will be more properly discussed in the numerical and conclusion sections.

---

> > ### Comment · Reviewer_USU8 · 2023-08-21
> >
> > I appreciate the additional discussions on k steps. Please add the neural network into discussion as well. With that, I look forward to having this paper accepted.

---

### Official Review · Reviewer_G3kk · 2023-07-09

**Soundness:** 4 excellent
**Presentation:** 3 good
**Contribution:** 3 good
**Rating:** 8
**Confidence:** 5

**Summary:**

This paper develops a convergence-rate analysis for the approximation of the Jacobian under the technique of one-step differentiation. This technique is similar to that of iterative differentiation (aka unrolling, differentiating through optimization), except only the last iterate of the algorithm is used to compute the Jacobian.

The authors develop a quite generic framework that contains many common algorithms as special cases. The paper also contains several examples of how to apply this framework to specific examples such as gradient descent and Newton's method.

**Strengths:**

1. The paper is clearly written, notation is largely standard. Main results are clearly explained. Overall, this paper was a pleasure to read.

2. As far as I can tell, the results in this paper are sound.

3. Experiments are to the point and clearly illustrate the theoretical results of the paper.

4. The topic considered is an important one. The authors take a technique that has been shown to work empirically and develop a sound theory around it.

**Weaknesses:**

1. The current framework doesn't apply to algorithms that either i) depend on past iterates, such as gradient descent with moment, or that ii) are not always contractive such as accelerated gradient descent.

It could be possible to easily extend the results to ii) by considering an enlarged space where x now contains the current iterate and past information (such as momentum), although I haven't checked whether its possible to get back the bound on the Jacobian after this transformation.

2. One of the most interesting results IMO is the convergence of a bi-level scheme, but the analysis done in 3.3 seems more of an afterthought. To start, the statement is highly confusing: its presented in terms of f, but the f here is not (if I understood correctly) the f of the inner optimization. Instead, it should be understood as any f that is L Lipschitz. In fact, (and this is nowhere written), to obtain a bound on the suboptimality of the bilevel problem, we should take f = g. Even with this in mind, casting the result of Lemma 3 into a bound for the bilevel problem requires to make a number of substitutions. Why not state clearly the rate for the bilevel problem in term of the quantities and constants that are defined for this problem?

**Questions:**

L20: I didn't understand the point of the phrase "it does not solely relies on the compositional rules of differential calculus" (and it has at least a grammatical error relies -> rely)

**Limitations:**

In general the paper is quite honest about its limitations. However, I suggest the authors expand a bit more on which class of algorithms verify and which don't Assumption 1 (see Weaknesses)

---

> ### Author Rebuttal · Authors · 2023-08-04
>
> We thank the referee for his feedback. We provide below detailed answers.
>
> 1. We agree with the referee that some algorithms may appear not directly in the form of a contraction or even not directly in the iterative form given in (1). A typical example is the heavy ball method for strongly convex problems: it should be considered in the phase space (because it is a second-order algorithm), and iterations are not contractive. However, the composition of a certain number of iterations is contractive in the phase space. We will add a remark and cite this example with a bibliographic reference to illustrate it.
>
> 2. We agree that Section 3.3 needs rewriting, and we will work on it in a revised version. More precisely, we will add a corollary for the bilevel problem by combining the results of Corollary 4 and Lemma 3. Furthermore, we will modify the notations of Lemma 3 as it is not harmonized with the rest of the text (x and f do not have the same meaning as above). We prefer to keep Lemma 3 as a separate result since it applies beyond bi-level problems.

---

> > ### Comment · Reviewer_G3kk · 2023-08-20
> >
> > I thank the authors for their answers. I keep my score unchanged.

---

### Official Review · Reviewer_WGTU · 2023-07-15

**Soundness:** 3 good
**Presentation:** 4 excellent
**Contribution:** 2 fair
**Rating:** 5
**Confidence:** 4

**Summary:**

This paper studies the gradient of iterative algorithms, specifically examining one-step differentiation of these algorithms. This approach replaces the complex Jacobian inverse in implicit differentiation with a simple and fast identity approximation. The work refines the theoretical approximation analysis for one-step differentiation and characterizes its efficiency under fast convergent algorithms. Numerical examples are provided to illustrate the well-foundness of the one-step estimator.


**Strengths:**

- Organization. The paper is well-structured overall. Each section has clear motivation, and there is good coherence in the transition of analysis. The presentation of the algorithms and analysis is easy to follow.
- Detailed approximation analysis. The work includes the approximation analysis for different algorithms under the one-step setting.

**Weaknesses:**

- Missing references. The work [1] also discusses the theoretical aspects of the differentiation approximation of iterative processes and should be included.
- Notations. The paper could benefit from more explicit notation to enhance readability.
- Novelty. While the novelty of the algorithm proposal is not a major concern, it becomes a considerable issue regarding the analysis part. The concept of one-step differentiation has been proposed for a while. Although this paper does not claim novelty in the algorithm proposal and instead focuses on the theoretical aspects, the approximation analysis is not sufficiently novel from a theoretical standpoint. I would expect more insights related to optimization and generalization when applying the one-step differentiation to learning problems.

[1] Zhengyang Geng, Xin-Yu Zhang, Shaojie Bai, Yisen Wang, and Zhouchen Lin. On training implicit models. Advances in Neural Information Processing Systems, 2021

**Questions:**

Please see above.

**Limitations:**

No limitations are included.

---

> ### Author Rebuttal · Authors · 2023-08-04
>
> We greatly thank the referee for his constructive comments. Regarding the weaknesses pointed out by the reviewer, we propose the following modification which we hope will address the concerns of the reviewers so that the reviewer can update his evaluation.
>
> - One-step differentiation has indeed connections with phantom gradients proposed in [1], we will definitely add and discuss the missing reference.
>
> - We will sharpen the notations throughout the text and propose a deep rewriting of Section 3.3 to avoid notational confusion.
>
> - Following the reviewer's comment, as well as the comments of other reviewers, we will rework Section 3.3 and add a corollary providing explicit optimization guarantees for the bilevel problem. This will combine Corollary 4 and Lemma 3. This constitutes a specific instantiation of our results for learning problems which are formulated as bilevel problems with applications in hyper-parameter tuning or meta-learning.

---

> > ### Comment · Reviewer_WGTU · 2023-08-19
> > **Reply to Authors**
> >
> > Thank you for your clarification! I'd keep positive toward acceptance.

---

### Author Rebuttal · Authors · 2023-08-04

We thank the referees for the feedback on our work. We are glad that the referees found our paper *well-written* (WGTU, G3kk, USU8, qFET) with a *clear message* (qFET). Both the theoretical analysis and our experiments seems to have been appreciated by the reviewers.

We propose to include remarks and discussions following their comments. We will also modify section 3.3: use different notations for Lemma 3 and add a corollary dedicated to bilevel problems (following remarks of WGTU, G3kk and qFET). We will also include more details on the numerical experiments (implicit differentiation implementation details and random generation details, following the remarks of agCr and qFET). We provide below a more detailed response to each comment of the reviewers.

---

### Decision · Program_Chairs · 2023-09-21

**Decision:**

Accept (spotlight)

**Comment:**

Theory and experiments complement one another well in this paper, which tackles automatically differentiating iterative algorithms (consider differentiating through the iterative optimization procedure, but using only the final iterate). All reviewers here recommend acceptance with positive reviews (some strongly so, and with high confidence!), many highlighting that it is both well presented and technically solid (with a detailed convergence analysis). During discussion, one reviewer underscored that the paper also synthesizes related work well.

The discussion with reviewers was productive. I will highlight especially the thread with Reviewer G3kk, which led to some clarifying remarks on heavy ball methods and contractivity. The exchanges with several reviewers also led to technical revisions and improvements in Section 3.3 that I agree with, including stating guarantees for the bilevel problem, as well as several new references to discuss, clarifications on the "K-step" variant of the estimator, and more. I recommend indeed following through on all of these edits, as the authors have indicated they will. That will strengthen the paper further.